# Effects of Drying Methods on Morphological Characteristics, Metabolite Content, and Antioxidant Capacity of *Cordyceps sinensis*

**DOI:** 10.3390/foods13111639

**Published:** 2024-05-24

**Authors:** Mengjun Xiao, Tao Wang, Chuyu Tang, Min He, Yuling Li, Xiuzhang Li

**Affiliations:** State Key Laboratory of Plateau Ecology and Agriculture, Qinghai University, Xining 810016, China; 15574237597@163.com (M.X.); 13085500761@163.com (T.W.); chuyutang0410@163.com (C.T.); himi1228@163.com (M.H.); xiuzhang11@163.com (X.L.)

**Keywords:** *Cordyceps sinensis*, drying method, quality, metabolomics, antioxidant ability

## Abstract

*Cordyceps sinensis* is a rare and endangered medicinal herb in China and a typical medicinal and food plant. Most of the research related to *Cordyceps sinensis* focuses on its pharmacological effects, artificial cultivation and clinical applications. However, there are few comprehensive evaluations on the quality of *Cordyceps sinensis* under different drying methods. In this study, the effects of vacuum freeze-drying (DG), oven-drying (HG) and air-drying (YG) on the morphological characteristics, microstructure, antioxidant activity and metabolites of *Cordyceps sinensis* were investigated using wild *Cordyceps sinensis* as the research object. The results showed that in their appearance and morphology, the YG- and HG-method *Cordyceps sinensis* samples were darker in color and wilted, while the DG- method *Cordyceps sinensis* samples were golden yellow in color and had better fullness. In terms of microstructure, the stomata of the YG and HG method *Cordyceps sinensis* samples were relatively small and irregularly shaped, whereas those of the DG method *Cordyceps sinensis* samples were larger and neat. In terms of antioxidant capacity, the HG-method samples were the lowest, followed by the YG group, and the DG group had the highest total antioxidant capacity. A correlation analysis revealed a significant relationship between antioxidant capacity and lipids, lipid molecules, nucleosides, nucleotides, and analogs. A metabolomics analysis identified 1937 metabolites from 18 superclasses, with lipids, lipid-like molecules, organic acids and derivatives, organoheterocyclic compounds, and organic oxygen compounds being the predominant metabolites in *Cordyceps sinensis*. Differentially accumulated metabolites (DAMs) in DG samples showed higher levels of lipids and lipid molecules, organic oxygen compounds, organic acids and derivatives, and organoheterocyclic compounds compared to the other drying methods, suggesting DG as the optimal preservation method for *Cordyceps sinensis*. These findings offer insights for selecting appropriate drying methods and maintaining the post-drying quality of *Cordyceps sinensis*.

## 1. Introduction

*Cordyceps sinensis*, a well-known medicinal fungus and traditional Chinese herbal medicine, is highly esteemed internationally for its healing and health benefits [1]. This fungus is primarily found in the Tibetan Plateau and surrounding regions, including Qinghai, Tibet, Gansu, Sichuan, and Yunnan provinces in China, as well as certain areas in Bhutan and Nepal, typically at altitudes ranging from 3500 to 5000 m [2]. Modern pharmacological studies have revealed its diverse biological activities, such as hepatorenal protection has, for example revealed its diverse biological activities; hepatorenal protection [3,4], immunomodulation [5], anti-aging [6], anti-inflammatory [7], and anti-tumor effects [8]. However, due to its highwater content, a suitable drying method is essential to preserve the nutritional value, efficacy, and taste of *Cordyceps sinensis* [9,10].

Drying plays a crucial role in the pharmaceutical process of traditional Chinese medicine (TCM), directly impacting drug quality [11]. Historically, Chinese herbal medicines have mainly been air-dried (YG), a cost-effective method with a long-standing tradition in our country. However, air-drying poses challenges such as extended drying times, significant loss of active ingredients, and susceptibility to mold [12]. With advancements in society and economy, oven-drying (HG) and vacuum freeze-drying (DG) have emerged as prevalent drying methods for food and pharmaceuticals [13,14,15]. HG involves electric heating for blast cycle drying, circulating hot air through a fan to maintain a uniform temperature. This method is known for its ease of operation, controllability, and high capacity [16]. On the other hand, DG represents a novel drying approach, conducted in a high-vacuum environment (below 609.3 Pa), where material water freezes and sublimates into vapor. Microbial reactions are inhibited at low temperatures and under vacuum, enhancing preservation of bioactive components and ensuring superior quality [17,18]. The current research on *C. sinensis* primarily focuses on its active ingredients, clinical applications, and artificial cultivation [19,20]. However, there is limited literature on the effects of metabolomics-based approaches on the quality of *C. sinensis* using different drying methods.

Metabolomics, a rapidly advancing technology, has significantly contributed to plant-related sciences and drug discovery [21]. Non-targeted metabolomics analysis, utilizing high-resolution mass spectrometry, allows for the unbiased measurement of metabolite [22,23]. Previous studies have explored metabolic changes in *Salvia miltiorrhiza* during post-harvest processing using a metabolomics approach to identify the optimal processing method [24]. Another study showed the changes in flavor quality and bioactive components of *Phlebopus portentosus* under low-temperature storage conditions, revealing the effect of the storage process on post-harvest quality [25]. Compared to traditional detection methods, metabolomics offers a more comprehensive and intuitive analysis of differences and similarities in active ingredients, providing a robust platform for evaluating product quality.

In this study, fresh *C. sinensis* was utilized as the primary focus to examine the impact of various drying methods on the morphological characteristics and antioxidant activity of *C. sinensis*. A non-targeted metabolomic analysis was employed to identify the metabolites present in wild *C. sinensis* and assess any alterations in its metabolite composition. These findings are valuable for selecting an appropriate drying method for *C. sinensis*, as well as for expanding the potential applications of dried *C. sinensis* and establishing drying standards within the food and medical sectors.

## 2. Materials and Methods

### 2.1. Sample Collection

All samples (CK, DG, HG, and YG; 6 repetitions) were purchased from Qinghai Baohuitang Bio-Technology Co., LTD, located in Xining City, Qinghai Province, China. Fresh *C. sinensis* samples were specifically collected in June 2023 from Maqin County (95°38′12″ E, 33°8′15″ N, H:4103), Guoluo Tibetan Autonomous Prefecture, China. Only samples that were in good shape and complete were chosen for experimentation, and all the samples were stored at −80 °C for backup purposes.

Ten grams of fresh *C. sinensis* (CK) were dried using three different methods: vacuum freeze-drying (DG), oven-drying (HG), and air-drying (YG). Drying was carried out until a constant weight was achieved. In the YG method, fresh *C. sinensis* was placed flat in a cool, dry area with a controlled ambient temperature (20–25 °C) and relative humidity (50–65%) until it reached a constant weight. For the HG method, fresh *C. sinensis* was evenly spread on trays in an electrically heated thermostatic drying oven (PH030A, Yiheng Technology Co., Ltd., Shanghai, China) at 60 °C with hot air until a constant weight was achieved, after which the samples were removed. In the DG method, fresh *C. sinensis* was spread out on trays without overlapping and freeze-dried in a vacuum freeze-dryer (Pilot 5-8S, Blocool Experimental Instruments Co., Ltd., Beijing, China) at −35 °C for 3 h of pre-cooling followed by 25 h of low-temperature, high-vacuum freeze-drying until a constant weight was achieved. The moisture content was then analyzed using the gravimetric method.

### 2.2. Appearance and Flavor Evaluation Analysis

The appearance and color of *C. sinensis* play a crucial role in assessing its freshness and quality. Sensory evaluation, which involves assessing product characteristics through sight, smell, and touch, is an effective method for evaluating the quality of *C. sinensis* [26,27]. By comprehensively examining the sensory attributes of *C. sinensis*, this method can provide a timely and accurate assessment of its quality, i.e., whether it has anomalies or not. Furthermore, this intuitive and simple approach offers a practical means of evaluating *C. sinensis*.

Scanning electron microscopy (SEM, Hitachi SU8100, Tokyo, Japan) was utilized to examine the microstructures of both the fresh and dried *C. sinensis* samples. Observations were conducted at magnifications of 200×, 500×, and 3000× with a voltage of 5.0 kV. Each group consisted of three replicates, and data analysis was performed using IBM SPSS Statistics software (version 26.0). Data visualization was performed using Origin 2022 software. Experimental results were presented as the mean ± standard error (SEM).

### 2.3. Antioxidant Activities Assays

#### 2.3.1. Measurement of Free Radical Scavenging Capacity

It is common to utilize DPPH, hydroxyl (OH^−^), and superoxide anion (O^2−^) free radical scavenging rate capacities to monitor changes in antioxidant activity over time. As antioxidant molecules react with radicals, the absorbance of the system decreases, with the rate of absorbance change serving as an indicator of antioxidant capacity [28,29]. The kits used in this study were purchased from Suzhou Keming Biotechnology Co. Ltd. (Shanghai, China), and the test methods followed the instructions provided with the kits and relevant references [30,31,32].

The DPPH free radical scavenging capacity absorbance was determined at 515 nm. The DPPH free radical scavenging capacities were calculated as by the following equation:SA_DPPH_ (100%) = [(A0 − Ai)/A0] × 100%
where A0: blank value and Ai: measured value.

The OH^−^ free radical scavenging activity absorbance was deter-mined at 550 nm. The OH^−^ assays were calculated by the following equation:SA _OH−_ (100%) = (A1 − Ai)/(A1 − A0) × 100%
where A1: control value A0: blank value, and Ai: measured value.

The O^2−^ radical scavenging capacity was determined by measuring the absorbance at 530 nm using an unsampled O2 scavenger solution as a blank.
SA _O−_ (100%)= [(A0 − Ai)/A0]× 100%
where A0: blank value and Ai: measured value.

#### 2.3.2. Determination of Total Antioxidant Capacity

The FRAP assay, is presented as a novel method for assessing “antioxidant power.” The FRAP assay is inexpensive, its reagents are simple to prepare, and the results it produces are highly reproducible [33].

The FRAP free radical scavenging capacity absorbance was determined at 593 nm. The FRAP free radical scavenging capacities were calculated as by the following equation:SA_FRAP_ = 0.4027× [(Ai − A0/W]
where A0: blank value, Ai: measured value, and W: sample mass (g)

### 2.4. Metabolite Identification and LC-MS Data Analysis

#### 2.4.1. Sample Preparation

Methanol, acetonitrile, ultrapure water, acetone, and formic acid (LC-MS grade) were obtained from Thermo Fisher Co., Ltd. (Shanghai, China). Other reagents with chemical grades were obtained from Beijing Chemical Reagents Company (Beijing, China). The samples were accurately weighed (100 ± 5 mg) and placed into 2 mL EP tubes along with 6 mm diameter grinding beads. A total of 400 µL of 80% aqueous methanol containing four internal standards, such as L-2-chlorophenylalanine, were added to the tubes. The samples were then ground for 6 min at −10 °C and 50 Hz using a frozen tissue grinder. Extraction was carried out through low-temperature sonication for 30 min at 5 °C and 500 kHz, followed by leave at −20 °C for 30 min. After centrifugation at 13,000 g for 15 min at 4 °C, the supernatant was pipetted for LC-MS analysis. Finally, 20 µL of supernatant was extracted from each sample and mixed to create QC samples.

#### 2.4.2. LC-MS Detection

The instrument platform for this LC-MS analysis was an ultra-high performance liquid chromatography tandem fourier transform mass spectrometry UHPLC -Q Exactive HF-X system (Thermo Fisher Scientific, Shanghai, China). The chromatographic conditions are as follows: ACQUITY UPLC HSS T3. Mobile phase A consisted of 95% water and 5% acetonitrile with 0.1% formic acid, while mobile phase B comprised 47.5% acetonitrile, 47.5% isopropanol, and 5% water with 0.1% formic acid. The injection volume was 3 μL, and the column temperature was maintained at 40 °C. Mass spectrometry conditions involved electrospray ionization for sample ionization, with mass spectrometry signals collected using a turbojet ion source. The heating temperature was set at 425 °C, the capillary temperature at 325 °C, and the ion injection voltage at 3500 V (POS)/−3500 V (NEG).

#### 2.4.3. Data and Statistical Analysis

The raw data were imported into the metabolomics processing software Progenesis QI (Waters Corporation, Milford, CT, USA) for baseline filtering, peak identification, integration, etc., resulting in a data matrix with information on retention times, mass-to-charge ratios, and peak intensities. Following the acquisition of metabolite spectral analysis data from the samples of the different methods, all peaks in the mass spectra of the substances were integrated based on peak area. The mass spectra of the same metabolite in the various method samples were then corrected by integration. Multivariate statistical analysis was conducted on the measured metabolomic data. Differential metabolites were identified in *C. sinensis* samples throughout the drying process based on specific criteria: a threshold variable importance in projection (VIP) value ≥ 1 and a significance level of *p* < 0.05.

Initially, principal component analysis (PCA) was used to gain insight into overall metabolite differences and sample variations. Subsequently, an orthogonal partial least squares discriminant analysis (OPLS-DA) was performed, with the model’s reliability assessed through 200 random permutation tests. A mantel test was utilized to examine the correlation between the identified differential metabolites and antioxidant capacity during drying. Hierarchical cluster analysis (HCA) was employed to visualize dynamic metabolite changes during drying using a heatmap.

## 3. Results

### 3.1. Effect of Different Drying Methods on the Morphological Characteristics of C. sinensis

The evaluation of the *C. sinensis* samples revealed that CK *C. sinensis* appeared golden-yellow, DG *C. sinensis* was brownish-yellow, while the HG and YG *C. sinensis* had a darker, sepia color. The DG *C. sinensis* had a light mushroom aroma, whereas both the YG and HG *C. sinensis* exhibited a strong mushroom scent. In terms of form and texture, the DG samples showed the best fullness of the worm body and cysts, maintaining the original morphology of *C. sinensis* pre-freeze-drying with normal cyst and worm body organization (Figure 1A–D).

Conversely, the HG and YG *C. sinensis* samples displayed a contraction in body volume, bending, and a dense, hard texture with increased chewing effort.

Scanning electron microscopy (SEM) is used for point-by-point imaging to capture magnified images [34]. The SEM images illustrate the microstructure of *C. sinensis* in both fresh and dried samples using three different methods. A comparison highlights significant variations in the microstructure of *C. sinensis* across the drying methods. The CK sample exhibited a dense structure with relatively denser pore channels and a more uniform channel clearance distribution (Figure 1A1–A3). In contrast, the DG samples showed relatively small pore channels and densely distributed pores, with some regions having fewer pores (Figure 1B1,B3). The HG-and YG-method samples displayed a dense reticulated structure with large pore openings, along with some mycelia clustered irregularly (Figure 1C1–D3).

### 3.2. Effect of Different Drying Methods on the Antioxidant Activity of C. sinensis

*C. sinensis* exhibits notable pharmacological effects, primarily attributed to its antioxidant properties [35,36]. This study evaluated the impact of various drying methods on the antioxidant capabilities of *C. sinensis* by measuring DPPH, OH^−^, and O^2−^ free radical scavenging rates, as well as FRAP total antioxidant capacity (Figure 2).

The DPPH radical scavenging capacity did not show significant differences (*p* < 0.05) between CK and DG groups but was significantly different (*p* < 0.05) from HG and YG groups. The hydroxyl radical scavenging capacity varied among the groups, with the DG group showing the second highest capacity after the CK group (up to 55%). The superoxide anion radical scavenging capacity differed significantly among CK, DG, HG, and YG groups, with that of the YG group being slightly higher than the HG group. The total antioxidant capacity also showed significant differences among the four groups. The DG samples exhibited significantly higher antioxidant capacity compared to other drying methods (*p* < 0.05), while the HG samples had the lowest total antioxidant capacity.

### 3.3. Metabolite Profiles

The quality control (QC) samples are prepared by mixing the extracts of all the samples in equal volume and are used in the same way for analyzing the samples. In the process of instrumental analysis, one QC sample is inserted into every 5–15 analyzed samples to examine the stability of the whole process of detection. The QC samples, were prepared by pooling equal aliquots of each drying method and fresh *C. sinensis* samples were examined to confirm the repeatability and reliability of the LC-MS method. The typical base peak chromatograms (BPCs) of the five QC samples were superimposed well, indicating consistent retention time and peak intensity (Appendix A). Additionally, 95.0% and 97.1% of features showed a relative standard deviation (RSD, %) < 30% in positive and negative ESI modes, respectively, confirming the reliability of the data (Appendix A). After filtering peaks with poor repeatability, 4179 and 4881 ion features were obtained from all *C. sinensis* samples in positive and negative ESI modes.

Using self-built and public databases, 1937 metabolites (Appendix A) were tentatively identified and classified into 18 categories, including lipids and lipid-like molecules, organic acids, organoheterocyclic compounds, and benzenoids (Figure 3A). Differential expression analysis of the nine superclasses in the four sample groups revealed that fresh *C. sinensis* had higher levels of lipids and lipid-like molecules, while dried *C. sinensis* contained more organic oxygen compounds, phenylpropanoids, and polyketides (Figure 3B).

### 3.4. Antioxidant Capacity and Correlation Analysis

The correlation between antioxidant capacity and various metabolites of *C. sinensis* was analyzed under different drying methods (Figure 4). Among the 14 classes of compounds studied, lipids and lipid molecules, nucleosides, nucleotides, and analogs exhibited significant correlations with DPPH, OH^−^, and O_2_^−^ free radical scavenging rates, as well as FRAP total antioxidant capacity. The antioxidant activity of *C. sinensis* was found to vary based on the drying method employed and was associated with the content in specific compounds.

### 3.5. Screening of Differentially Accumulated Metabolites (DAMs) and Hierarchical Cluster Analysis (HCA)

PCA, an unsupervised pattern recognition method, was utilized to visualize the general clustering trends of different groups and the degree of variability of samples within the same group. The results of PCA analysis showed that all samples can be divided into four different groups PC1 (55.4%) and PC2 (17.2%) (Figure 5A). Six biological replicates from each of the four groups were closely clustered together, confirming the consistency and reliability of the LC-MS method. It is worth noting that PCA analysis is limited in its ability to eliminate between-group and random errors and maintain the integrity of the data. On the other hand, OPLS-DA, a supervised discriminant analysis statistical method, provides a more comprehensive approach to identifying differences between groups [37]. The OPLS-DA score plot effectively distinguished between different drying methods of *C. sinensis*, aligning with the findings of the PCA analysis. Furthermore, the high values of R2X, R2Y, and Q2 in the OPLS-DA model indicated its reliability in classifying freshly harvested and dried *C. sinensis*, showcasing its stability, reliability, and predictive capabilities (Appendix A).

The screening criteria for significant differentially abundant metabolites (DAMs) were VIP ≥ 1 and *p* < 0.5. Venn diagrams, volcano plots, and DAMs were utilized to visualize the similarities and differences in metabolites across the four sample groups. It was identified that each drying method exhibited its own distinct set of metabolites, with 43 DAMs in YG vs. HG samples and 86 DAMs in DG vs. HG samples (Figure 5B). The DG samples showed higher levels of organic oxygen compounds, organic acids and derivatives, and organoheterocyclic compounds. Moreover, 75 common DAMs were identified as potential biomarkers for distinguishing among between the three methods (Appendix A).

The hierarchical clustering analysis (HCA) based on the top 25 metabolites revealed that different metabolites were grouped together, suggesting similar expression patterns among the samples. The clustering results indicated variations in gene expression among the samples. Differential metabolites are visually represented in the volcano plot (Appendix A). A total of 495 metabolites were found to be upregulated and 263 downregulated between the DG and CK samples, 475 upregulated and 273 downregulated between the HG and CK samples, 355 upregulated and 352 downregulated between the YG and CK samples, 368 upregulated and 195 downregulated between the DG and HG samples, 430 upregulated and 114 downregulated the between DG and YG samples, and 117 upregulated and 476 downregulated between the HG and YG samples.

The HCA results indicated that the differential metabolites formed two distinct groups. Group 1 consisted of three metabolites with an overall upregulated trend, while Group 2 consisted of 22 metabolites with an overall downregulated trend. A significant number of differential metabolites, classified as lipids and lipid-like molecules, were notably upregulated in the DG samples. In comparison to the YG and HG samples, lipid metabolites such as 4-Hydroxyproline galactoside, 9,10-Dihydroxystearic acid, and LysoPC (16:0/0:0) exhibited significantly higher expression levels, while organic oxygen compounds like oleuropein, sucrose, and tazolol were notably upregulated in the YG samples. Overall, the variations in metabolites among the three different drying methods of *C. sinensis* were primarily observed in lipids and amino acids, offering valuable insights for identifying potential biomarkers for differentiation.

### 3.6. KEGG Pathway Enrichment Analysis

Based on the KEGG pathway database, this study annotated and enriched DAMs, revealing that functional pathways in the three control groups shared 10 metabolic pathways (Appendix A). Among these, four pathways were highly significant (*p* < 0.001): alpha-linolenic acid metabolism, nucleotide metabolism, glycerophospholipid metabolism, and linoleic acid metabolism (Figure 6A,C,E). The results of enrichment of metabolic pathways showed that the lipid levels of the four groups of samples changed most significantly, glycerophospholipid metabolism was a common significant enrichment pathway of the three comparison groups, and only one metabolite was an organic acid, and the rest were lipids, Glycerylphosphorylcholine, GPEtn (14:0/yonder, PC (17:0/0:0), etc. (Figure 6B,D,F). These results indicate that these pathways are significant targets for studying metabolite changes and quality formation in *C. sinensis* under different drying methods. In summary, alpha-linolenic acid metabolism, nucleotide metabolism, glycerophospholipid metabolism, linoleic acid metabolism, and the metabolic intensity of linoleic acid metabolism in these significant enrichment pathways, and the accumulation pattern of secondary metabolites are among the main factors affecting the quality of *C. sinensis* under different drying methods.

## 4. Discussion

Both the PCA and HCA cluster analyses indicated suggesting variations in the metabolic profiles between fresh and differently dried samples. From the analysis of their appearance, the three drying processes had obvious effects on the color, odor, and fullness of *C. sinensis*. The order of superiority of appearance quality was DG > YG > HG, indicating that the DG method could better maintain the appearance, morphology, and color of fresh products before drying. The DG-dried samples exhibited better resistance to color degradation. This is due to the lower oxygen content and temperature in the vacuum chamber, which inhibits the enzymatic browning reaction and pigment destruction [38]. DG showed higher hardness values for dried samples. This is associated with severe drying shrinkage [39,40]. The YG-and HG samples had a strong mushroom flavor, and the DG samples had a slight mushroom flavor. The YG and HG processes were conducted at relatively high temperatures, while the DG process was conducted at low temperatures, therefore, we speculate that temperature affects the odor of dried samples. Odors are caused by amino acids, organic acids, and phenolics, and higher temperatures accelerate the release of these substances [41]. Based on preliminary work, it was found that *C. sinensis* contains substances such as amino acids, organic acids, and nucleotides, which may contribute to its odor characteristics.

The analysis of the microstructure showed that there were significant differences in the microstructure of *C. sinensis* after the three different drying methods, and the DG and CK samples had larger stomata and more regular pore distributions with similar microstructures. This might be due to rapid freezing and vacuum conditions, when the frozen water molecules changed into the gaseous state by sublimation, leaving the pores in the inner section. Meanwhile, the precooling process of DG immobilizes the tissue structure of the samples, which leads to structural rigidity to maintain the original traits [42,43].

*C. sinensis* has remarkably beneficial pharmacological effects, most of which are associated with antioxidant capacity [44]. In this study, the values of DPPH, OH^−^, and O^2−^ free radical scavenging rate, and FRAP total antioxidant capacity of *C. sinensis* were determined using different drying methods. The results showed that the DG samples had excellent antioxidant capacity in all the dried samples, which may be because the prolonged heating method and exposure to oxygen during the drying process usually accelerated the degradation of the active ingredients responsible for the antioxidant capacity. Owing to the low temperature and pressure of the DG method, many compounds are unable to react with oxygen and, therefore, have high antioxidant capacities. Correlation analysis showed that lipids and lipid molecules, nucleosides, nucleotides, and analogs were significantly correlated with each other in antioxidant capacity. Therefore, the content of these two metabolites may affect the antioxidant capacity of *C. sinensis*. Related studies have shown that the aqueous extract of *C. sinensis* is rich in polysaccharides, five alditols (mainly mannitol), eight nucleosides, proteins, and polyphenols, and have demonstrated that these chemical constituents have a greater effect on the antioxidant capacity, which is consistent with the results of this study [45].

In this study, 18 categories of 1937 metabolites were identified, among which lipids and lipid molecules, organic acids and their derivatives, and organic heterocyclic compounds were the main active substances of *C. sinensis*. The analysis results of the earlier Venn diagram show that the DG and YG samples had the least differential accumulation of metabolites, suggesting a higher similarity in the types and quantities of metabolites in these two groups. Both DG and YG samples were exposed to relatively low temperatures, implying a potential influence of temperature on metabolites. The DG method demonstrated effectiveness in preserving metabolite integrity, possibly due to the reduction in enzyme activity at low temperatures, thereby safeguarding the metabolites from enzymatic degradation [46]. Among the significantly differentially abundant metabolites (DAMs), lipids and lipid-like molecules, organic acids and derivatives, organoheterocyclic compounds were the most abundant superclasses among the common DAMs, while phenylpropanoids and polyketides, organic nitrogen compounds, alkaloids, and derivatives were the least abundant. Lipids are a class of bioorganic molecules that are insoluble in water but soluble in non-polar solvents, and their structural diversity provides them with a variety of important physiological functions [47,48]. Lipids are more responsive to different methods (e.g., freeze-drying) and stress (e.g., oxidative damage). Therefore, changes in lipid levels can directly and accurately reflect the real-time physiological status of *C. sinensis* [49,50]. Lipids and lipid-like molecules are the main active components of *C. sinensis*, and the highest content of lipids and lipid-like molecules is found in the DG method; some examples of these lipids and lipid-like molecules are 5-Hexyl-2-furanoctanoic acid, Dexpanthenol, and 2-Hydroxyadipic acid. These results, on the one hand, provide a basis for further study of its pharmacologically active substances and on the other hand, provide theoretical support for the development of functional health food of *C. sinensis*.

## 5. Conclusions

In this study, the effects of different drying methods on the morphological characteristics, antioxidant activity, and metabolites of *C. sinensis* were compared. The results showed that the DG samples had similar morphological characteristics to the CK samples and possessed strong antioxidant activity. A total of 1937 metabolites were detected by metabolomics, and the different drying methods significantly altered the composition and content of *C. sinensis*. DG was able to maintain the integrity of the metabolites better, with a higher abundance of lipid and lipid-like molecules, organic acids and their derivatives, and organic heterocyclic compounds. So, this may be the reason why DG has been able to maintain better commercial performance. However, this highly complex equipment and the required energy consumption significantly increase production costs. In practical applications, a suitable drying method can be selected, considering the purpose of use and the process cost. On the one hand, this study provides a basis for further research on the medicinal effects of *C. sinensis*, and on the other hand, it also provides theoretical support for the development of functional health food products from *C. sinensis*.

## Figures and Tables

**Figure 1 foods-13-01639-f001:**
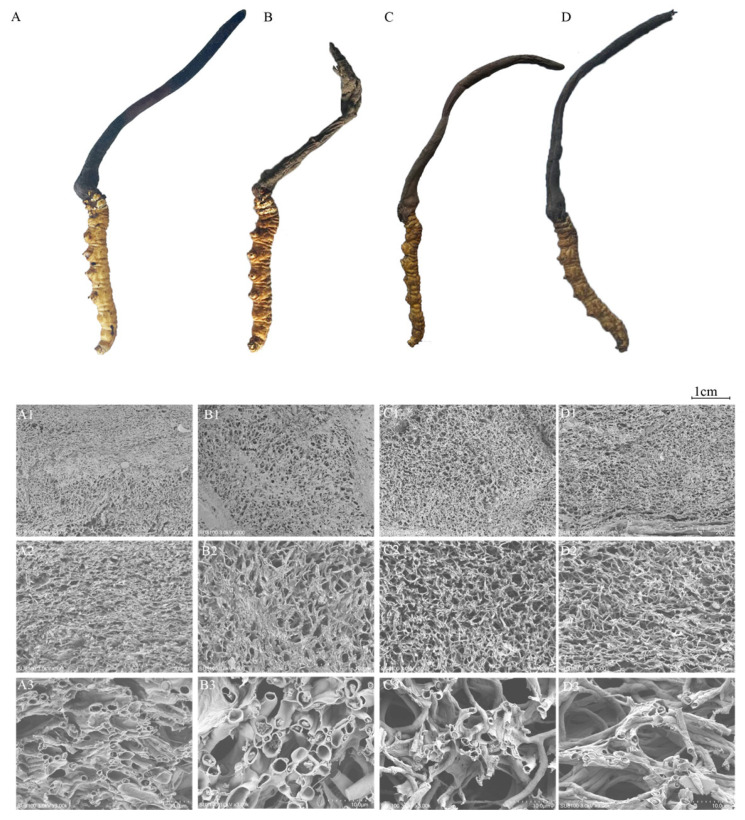
Total ion chromatograms of quality control (QC). Morphology and microstructure of *C. sinensis* samples subjected to different drying methods. (**A**) CK: Fresh *Cordyceps sinensis*, (**B**) DG: vacuum freeze-drying, (**C**) YG: air-drying, and (**D**) HG: oven-drying. Scalebar (**A1**–**D1**) 200× magnification, (**A2**–**D2**), 500× magnification, (**A3**–**D3**) 3000× magnification.

**Figure 2 foods-13-01639-f002:**
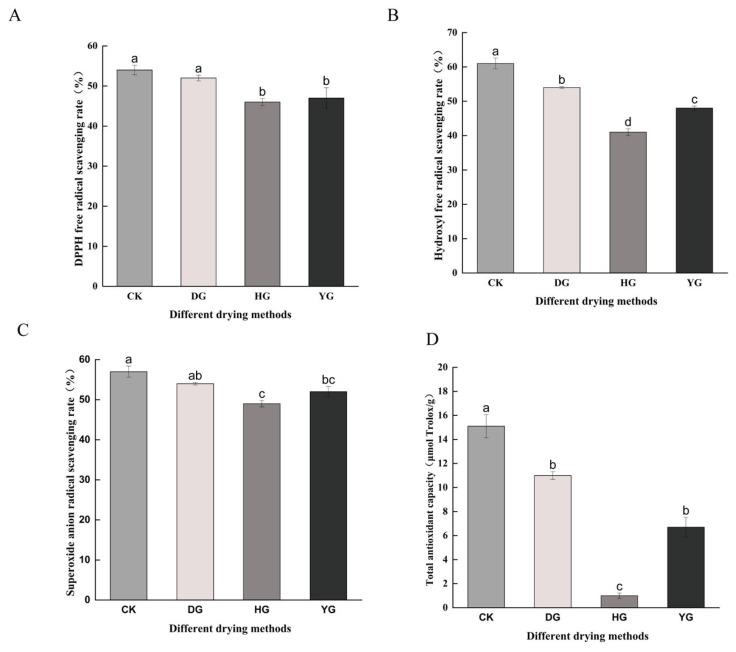
Antioxidant capacity under different drying methods: (**A**) DPPH (DPPH free radical scavenging rate), (**B**) OH^−^ (Hydroxyl free radical scavenging rate), (**C**) O^2−^ (Superoxide anion free radical scavenging rate), (**D**) FRAP (Total antioxidant capacity). Different letters represent significant differences between groups. Identical letters indicate that there is no significant difference between the two groups (*p* > 0.05), while different letters indicate that there is a significant difference between the two groups (*p* < 0.05).

**Figure 3 foods-13-01639-f003:**
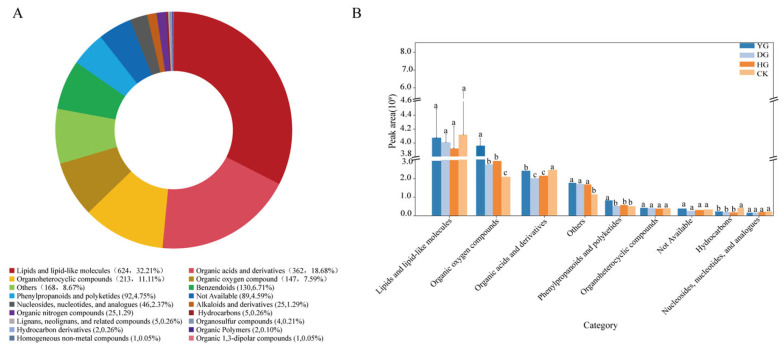
(**A**) Metabolomic profiles of CK (Fresh *Cordyceps sinensis*), DG (vacuum freeze-drying), YG (air-drying), and HG (oven-drying), (**B**) Expression of the top nine superclasses in the four groups’ samples. Different letters represent significant differences between groups. Identical letters indicate that there is no significant difference between the two groups (*p* > 0.05), Different letters indicate that there is a significant difference between the two groups (*p* < 0.05).

**Figure 4 foods-13-01639-f004:**
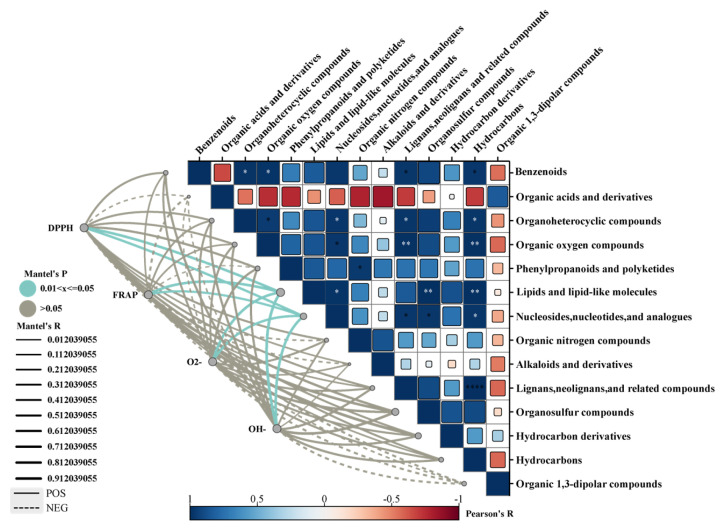
Correlation between antioxidant capacity and differential metabolites of *C. sinensis* under different drying methods (* *p* < 0.05; ** *p* < 0.01, **** *p* < 0.0001).

**Figure 5 foods-13-01639-f005:**
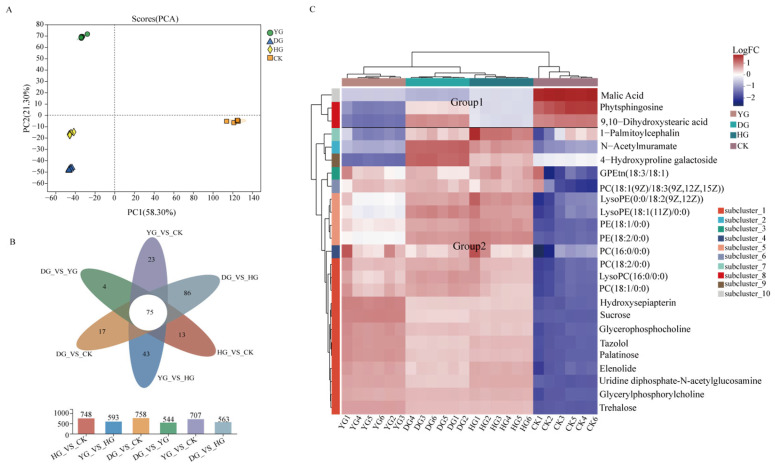
(**A**) Principal component analysis (PCA) results showing metabolite profile differences, (**B**) Venn plot of the number of differential metabolites among the three comparison groups, (**C**) HCA of DAMs in all samples.

**Figure 6 foods-13-01639-f006:**
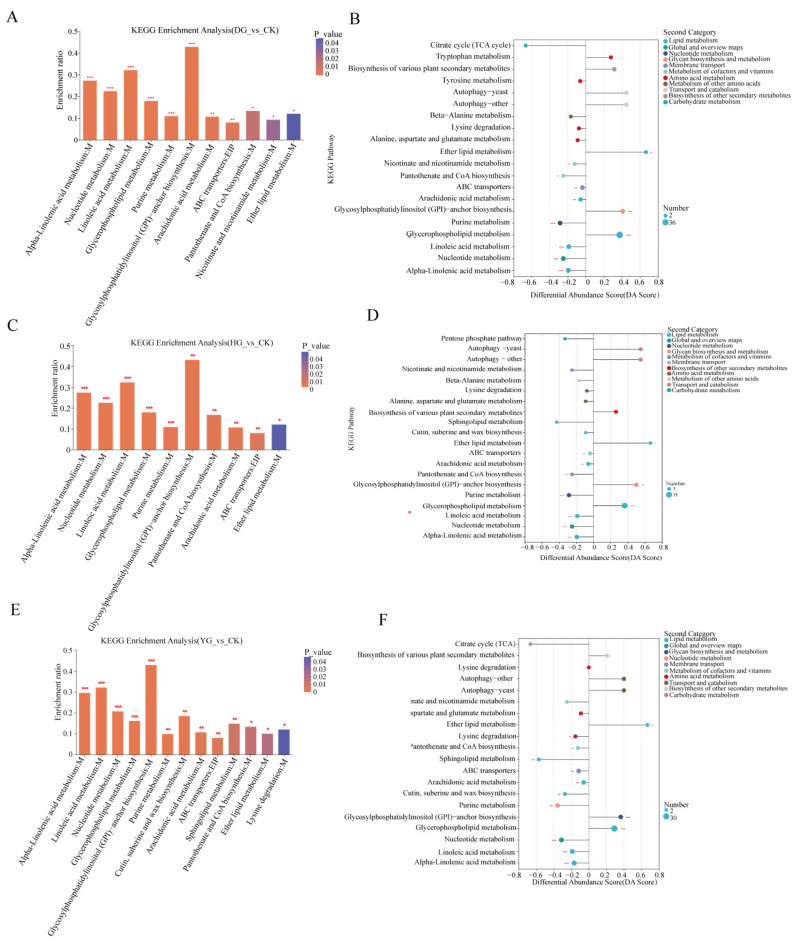
(**A**,**C**,**E**) KEGG enrichment analysis, (**B**,**D**,**F**) KEGG pathway differential abundance score maps. *p*-value or FDR < 0.001 is marked ***, *p*-value or FDR < 0.01 is marked **, and *p*-value or FDR < 0.05 is marked *. M: metabolism; EIP: environmental information processing.

## Data Availability

The original contributions presented in the study are included in the article and Appendix A, further inquiries can be directed to the corresponding author.

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
