# Peer review of "Effects of Drying Methods on Morphological Characteristics, Metabolite Content, and Antioxidant Capacity of Cordyceps sinensis"

_foods, 2024, doi:10.3390/foods13111639_

Round 1
Reviewer 1 Report
Comments and Suggestions for Authors
Dear Authors,
This manuscript is dealing with the research of application of different methods for drying the Cordyceps sinensis and determination of morphological characteristics and biological activity of samples.
Abstract is general. The most important results are missing. Insert it in abstract.
How did you choose metabolites from Cordyceps sinensis which were analyzed in this manuscript?
In the subsection 2.1. are missing important information regarding Cordyceps sinensis. Insert in manuscript answers on the following questions:
What was geographical origin of Cordyceps sinensis sample?
Did Cordyceps sinensis was obtained after artificial cultivation?
What climate conditions were in the area from which was obtained Cordyceps sinensis?
How different climate conditions affects on the cultivation?
Insert the name of producer, city and country for freeze dryer.
Insert the name of producer, city and country for all equipment which was used for analysis in this manuscript.
Insert following subtitle Antiradical and antioxidant activities assays for the subsection 2.3. FRAP method is antioxidant method and all three other are antiradical.
In the section 2.4 are missing conditions for the LC-MS analysis.
Insert the name of equipment which was used for LC-MS analysis.
Highlight in manuscript which external standards were used in the subsection 2.4.1. .
In the subsection 2.4.2. insert which equipment was used for analysis. Insert the name of producer, city and country.
In the title of section 2.4. is mentioned LC-MS analysis, but in the subsection 2.4.2. is mentioned UPLC. It is very confused. Correct it.
Explain abbreviation QC in the section 3.3.
At the beginning of page 10 in the manuscript do not use term speculate since it is not suitable for scientific writing which have to be precise.
Wish you all the best in the future work,
Reviewer 2 Report
Comments and Suggestions for Authors
Dear Authors,
The current paper is interesting and complex in terms of results and data interpretation. Some corrections are however, required to improve the clarity of the manuscript and data presentation.
Please find my suggestions and observations below.
In the abstract, please add more data from the paper.
In the material and methods section, is not clear how many repetitions for each sample were conducted. Please mention that.
Results section.
From this part, the authors should move significant amount of information in the discussion chapter. Please mention only the main findings, and the rest of it, can be moved in the discussion section, to enhance that part, which is quite scarce.
Under each figure, please explain the meaning of a,b,c and d letters. In figure 2, what is the unit measurement for abundance? Also, all the abbreviations such as Ck, DK, FRAP and so on, all of them must be explained under the figure.
Figure 3. The metabolites profile is barely visible in the legend. I suggest placing the figure A and B under each other instead of besides for better visualisation. To express the significance, please maintain the same notation, a.b.c and d or ***.
Section 3.5. is very long for a results section, it is suggested that part of the text to be moved in the discussion section.
Same observation for figure 5.
Also it is worth mentioning what is the practical application of the studied plant material. In what industry? For what purposes?
Reviewer 3 Report
Comments and Suggestions for Authors
The research is very interesting and necessary, there are few scientific works published so far describing the results of scientific research on cordyceps, which is used in traditional Chinese medicine as an adaptogen. Despite many years of tradition, the chemical composition and biological properties of cordyceps are poorly known. Hence, the work significantly increases our knowledge in this area. The authors studied the composition and properties of this mushroom depending on various drying methods. Although the final conclusion about the most effective method of drying, i.e. freeze-drying, is quite obvious, many analyzes have been performed to draw conclusions regarding the correlation between individual metabolites. The work is very well documented, the authors provide many details about the experimental work and attach additional materials with analysis results, but some information is missing, e.g. a brief description of the methods used in the DPPH, FRAP, O2- and OH- tests. The charts require significant improvement and are difficult to read. In Figure 2, there are no markings of the measured values and units. The remaining drawings require reorganization, the small font makes it impossible to read the drawings.
In the discussion, some statements regarding the function of lipids, being quite obvious, may be removed.
